# Diversified regulation of circadian clock gene expression following whole genome duplication

**Alexander C. West**[1], **Marianne Iversen**[1], **Even H. Jørgensen**[1], **Simen R. Sandve**[2], **David G. Hazlerigg**[1], **Shona H. Wood**[1] *

**1** Arctic chronobiology and physiology research group, Department of Arctic and Marine Biology, UiT – The Arctic University of Norway, Tromsø, Norway, **2** Centre for Integrative Genetics (CIGENE), Department of Animal and Aquacultural Sciences (IHA), Faculty of Life Sciences (BIOVIT), Norwegian University of Life Sciences (NMBU), Ås, Norway

* shona.wood@uit.no

**Data Availability Statement:** The relevant scripts, input files and trees for the evolutionary analysis are available here: https://gitlab.com/sandve-lab/salmon-clockgenes/-/tree/master/. Publicly

## Abstract

Across taxa, circadian control of physiology and behavior arises from cell-autonomous oscillations in gene expression, governed by a networks of so-called 'clock genes', collectively forming transcription-translation feedback loops. In modern vertebrates, these networks contain multiple copies of clock gene family members, which arose through whole genome duplication (WGD) events during evolutionary history. It remains unclear to what extent multiple copies of clock gene family members are functionally redundant or have allowed for functional diversification. We addressed this problem through an analysis of clock gene expression in the Atlantic salmon, a representative of the salmonids, a group which has undergone at least 4 rounds of WGD since the base of the vertebrate lineage, giving an unusually large complement of clock genes. By comparing expression patterns across multiple tissues, and during development, we present evidence for gene- and tissue-specific divergence in expression patterns, consistent with functional diversification of clock gene duplicates. In contrast to mammals, we found no evidence for coupling between cortisol and circadian gene expression, but cortisol mediated non-circadian regulated expression of a subset of clock genes in the salmon gill was evident. This regulation is linked to changes in gill function necessary for the transition from fresh- to sea-water in anadromous fish. Overall, this analysis emphasises the potential for a richly diversified clock gene network to serve a mixture of circadian and non-circadian functions in vertebrate groups with complex genomes.

## Author summary

The generation of daily (circadian) rhythms in behaviour and physiology depends on the activities of networks of so-called clock genes. In vertebrates, these have become highly complex due to a process known as whole genome duplication, which has occurred repeatedly during evolutionary history, giving rise to additional copies of key elements of

available data was used to assess the multi-tissue expression in the Atlantic salmon, these data can be found in the NCBI Sequence Read Archive (SRA): PRJNA72713 and PRJNA260929. All RNA-seq data for the smoltification experiment is available in the European nucleotide archive under project number: PRJEB34224. Nanostring data can be accessed on GEO under the project identifier GSE146530. Source data for the cortisol assay in Fig 4A is provided in S4 Table. Source data for the qPCR in Fig 5 are provided in S6 Table.

**Funding:** The work was supported by grants from the Tromsø forskningsstiftelse (TFS) starter grant TFS2016SW and the TFS infrastructure grant (IS3_17_SW) awarded to SHW. ACW is supported by the Tromsø forskningsstiftelse (TFS) grant awarded to DGH (TFS2016DH). Simen Sandve wishes to acknowledge the funding of the Norwegian Research Council project Rewired (NRC project number 274669). The authors also thank Dag Inge Våge and Thu-Hien To at the NMBU Elixir-node for making the orthogroups and gene trees available. Research council of Norway grant number 241016 and Sentinel North Transdisciplinary Research Program Université Laval and UiT awarded to DGH. The publication charges for this article have been part funded by a grant from the publication fund of UiT-The Arctic University of Norway. The funders had no role in study design, data collection and analysis, decision to publish, or preparation of the manuscript.

**Competing interests:** The authors have declared that no competing interests exist.

the clock gene network. It remains unclear whether this results in functional redundancy, or whether it has permitted new roles for clock genes to emerge. Here, based on studies in the Atlantic salmon, a species with an unusually large complement of clock genes, we present evidence in favour of the latter scenario. We observe marked tissue-specific, and developmentally-dependent differences in the expression patterns of duplicated copies of key clock genes, and we identify a subset of clock genes whose expression is associated with the physiological preparation to migrate to sea, but is independent of circadian regulation. Associated with this, cortisol secretion is uncoupled from circadian organisation, contrasting with the situation in mammals. Our results indicate that whole genome duplication has permitted clock genes to diversify into non-circadian functions, and raise interesting questions about the ubiquity of mammal-like coupling between circadian and endocrine function.

## Introduction

Circadian control of metabolic physiology and behaviour is a ubiquitous characteristic across taxa [1–3]. In eukaryotes, circadian control derives from a cell-autonomous molecular oscillator, assembled from a regulatory network of transcription factors, co-factors, (co-) regulators, chromatin modifiers and an array of post-translational regulators of protein function, often described collectively as 'clock genes' [1]. Clock gene oscillations coordinate the transcription of multiple genes to exert effects on global cell metabolism [1]. While the molecular clock is conserved between insects and mammals [2], the mammalian network contains many duplicated components as a consequence of both local and whole genome duplication (WGD).

Two rounds of WGD preceded the establishment of the tetrapod lineage 500 million years ago (MYA) (Fig 1A), and gave rise to the complement of clock genes seen in mammals, including multiple paralogues of *Period* and *Cryptochrome* genes. Paralogues arising from WGD are known as 'ohnologues', after Susumu Ohno, who wrote a seminal monograph hypothesising that the genetic redundancy proceeding WGD facilitates evolutionary innovation [4,5]. Nevertheless, the evolutionary importance and extent to which clock gene ohnologues are functionally divergent largely remains unclear [6–10]. Indeed the retention of multiple redundant ohnologues of core clock genes is puzzling given that the essential role of the circadian clock has not changed during the course of evolutionary history [1,2,11,12]. Conceivably, functional differences between ohnologues, achieved either by coding sequence differences or by promoter-based differences in expression level, could enable tissue-specific optimization of function, but evidence for this is sparse [11,12]. It has been suggested that preferential interactions of ancient duplicated mammalian PERIOD proteins with specific duplicated mammalian CRYPTOCHROME proteins may affect photic entrainment [13], but experimental evidence is lacking [14]. Tissue-specific functions of mammalian CKIδ/ε ohnologues in regulation of PERIOD protein stability have been suggested [15], and alterations in period (tau), amplitude and clock resetting behavior have been observed but clear distinctions of function between the ohnologues are lacking [8,16–18].

Following the two basal vertebrate WGD events (Fig 1A), subsequent rounds of WGD have occurred in several linages, resulting in highly complex genomes containing thousands of ohnologue pairs. This is exemplified by the situation found in the salmonids, which underwent two additional rounds of WGD compared to basal vertebrates; a third WGD (Ts3R) shared by all teleost fish, and a more recent, the salmonid-specific fourth round of duplication (Ss4R) taking place some 100 MYA (Fig 1A) [19]. The Ss4R event is a defining characteristic of the

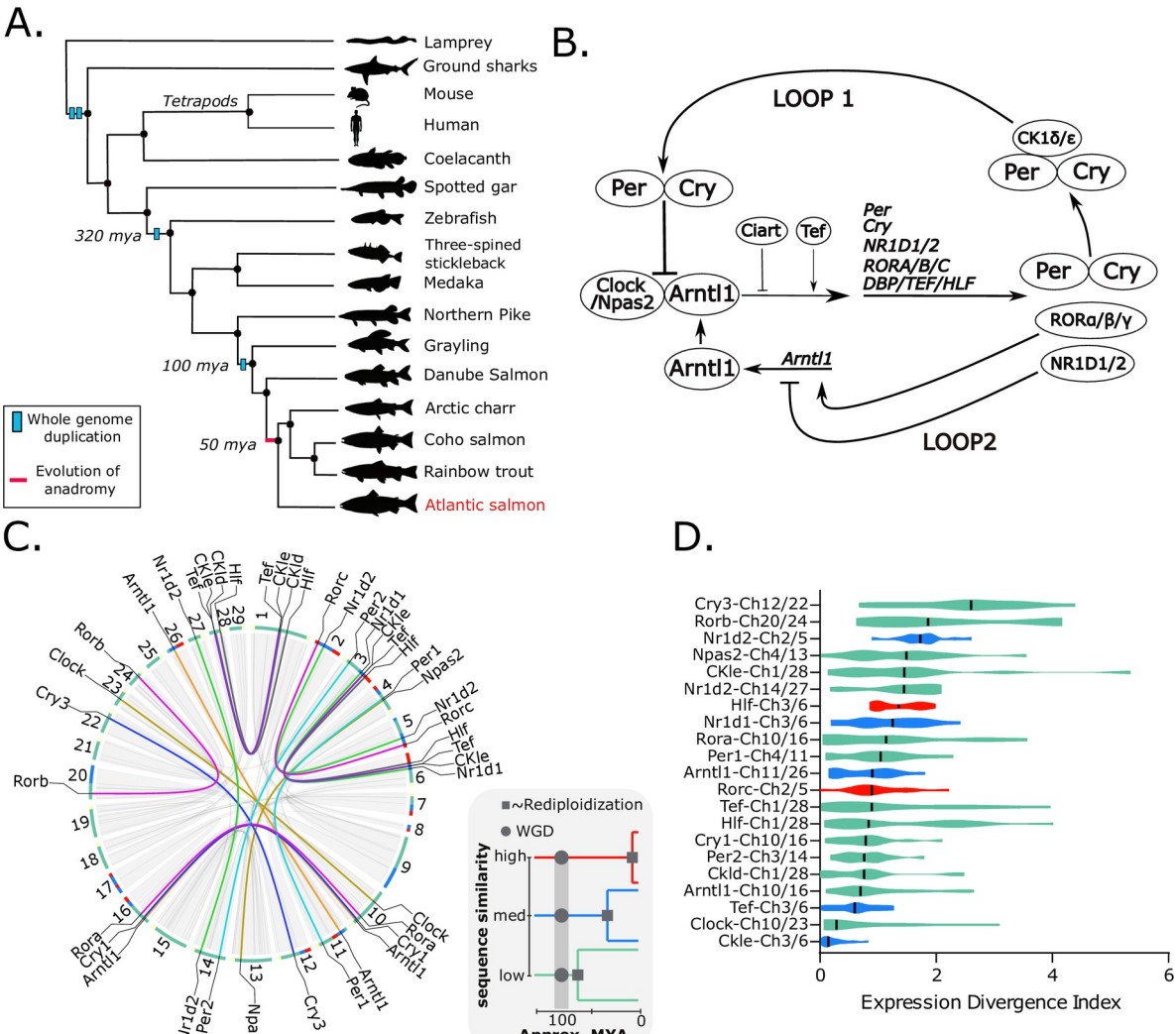

**Fig 1. Clock gene Ss4R duplicate pairs are divergently expressed in different tissues.** A. Phylogenetic tree highlighting WGD events and evolution of anadromy. All species represented (excluding the lamprey, ground sharks and coelacanth) were included in the phylogenetic identification of Atlantic salmon clock genes. B. The mammalian molecular clock network. C. Circos plot representing the Atlantic salmon chromosomes and the large collinear blocks of the genome that are duplicated (grey blocks/lines). Overlayed are the chromosomal positions of all Ss4R clock ohnologues, duplicated clock genes are connected by a coloured line. The colour of this line represents the different orthogroups. Around the outside of the circos plot sequence similarity of the loci in a 1mb window is shown as a coloured bar; high similarity >95% (red), medium 95–90% (blue), low ~87% (green). Sequence similarity on a genome wide level relates to approximate rediploidization time of Atlantic salmon chromosomes (see grey box). D. Differential regulation of Ss4R pairs in a panel of 11 different tissues. For each Ss4R pair, in each tissue, an expression divergence index (EDI) index was calculated (EDI = abs(log2[Gene1/Gene2])). The graph shows a violin plot of the distribution of EDI values across all tissues. The vertical black bar represents the median value. Approximate rediploidisation time of each pair is represented by a colour: red—late, blue—mid, green—early.

salmonid group and is theorized to have led to the evolution of anadromy; an adaptation of freshwater salmonids to spend part of their life-cycle at sea [20]. Genome-scale analysis in salmonids has begun to provide new insights into the evolutionary significance of ohnologue divergence [21,22]. Impressively, even though gene loss often occurs following duplication events (reviewed in: [23]), there remains a rich complexity of clock genes in teleosts compared to mammals. Of the 18 clock genes (as defined in Fig 1B) identified in laboratory mice the

zebrafish genome contains 30, and the Atlantic salmon genome contains 61 clock genes (S1 Table, S1 Appendix).

To understand why so many additional copies of core clock genes are retained, we have undertaken a comprehensive analysis of clock gene expression in the Atlantic salmon, exploring temporal regulation in different tissues and responsiveness to different environmental stimuli. Here, we show diversified regulation of clock ohnologues as a result of WGD, reflecting the fundamental differences in temporal organization of metabolism between tissues.

## Results

### Tissue-specific expression of clock gene ohnologues indicates regulatory divergence

To identify all conserved clock genes in the Atlantic salmon we extracted amino acid sequences from the well-characterized mouse clock gene network (Fig 1B) then searched for homologous sequences in Atlantic salmon [22] and 12 other vertebrates (S1 Appendix, Materials and methods, *Evolutionary analysis*). Homology relationships between protein sequences were traced back to the root of the vertebrate tree, revealing 61 canonical clock genes in the Atlantic salmon. Comparing the repertoire of clock genes in spotted gar (2 WGDs), zebrafish (3 WGDs), and Atlantic salmon (4 WGDs) (Fig 1A) we find no difference in gene retention for loop 1 versus loop 2 genes (Fig 1B). Forty-two of the 61 salmon circadian genes are duplicates arising from the salmonid specific genome duplication and can be assigned to 21 Ss4R specific ohnologue pairings (referred to as Ss4R pairs from here on), while for the remaining 19 genes no extant Ss4R duplicate can be identified suggesting gene loss after WGD (S1 Table, S1 Appendix). The chromosomal locations of the Ss4R pairs are shown on Fig 1C, and for ease of comparison the genes will be referred to by their gene name and chromosome. The S1 Table lists the specific gene identifiers and orthologues to zebrafish, medaka, spotted gar and mouse.

Following an autopolyploidization, such as the Ss4R, the tetraploid genome will accumulate mutations which block recombination and thereby accelerate duplicate divergence (referred to as rediploidization) [24]. This process has occurred at different rates in different genomic regions in salmonids. Using published data [22] on sequence similarity in 1Mbp windows across syntenic Ss4R regions we could classify the rediploidization times for our 21 Ss4R pairs from early (approx. 87% sequence similarity) to late (>95% sequence similarity) (Fig 1C) and assess whether the history of rediploidization was associated with regulatory divergence.

RNA profiling from 13 different tissues [22] demonstrated tissue-specificity of clock gene expression, and particularly highlighted the wide variety and high abundance of clock genes in the brain (S1 Fig). To assess the divergence between Ss4R pairs we calculated an expression divergence index (EDI), based on the relative expression of each member of a pair across all tissues expressed as a ratio (Fig 1D). This revealed evidence for divergent tissue-specific expression within multiple Ss4R pairs but no clear relationship to approximate time of rediploidization (Fig 1D). The Cry3-Ch12/Ch22 pair had the highest EDI, largely attributable to divergent expression in the brain and gill (S1 Fig). The three Ss4R pairs of *Nr1d1* (*Rev-erbα*) and *Nr1d2* (*Rev-erbβ*), which encode transcriptional repressors linking the circadian clock to energy metabolism [25], were also highly divergently expressed genes, again due to differences in the brain and gill (Fig 1D, S1 Fig). Hence tissue-specific expression divergence is a feature of particular aspects of the circadian clockwork.

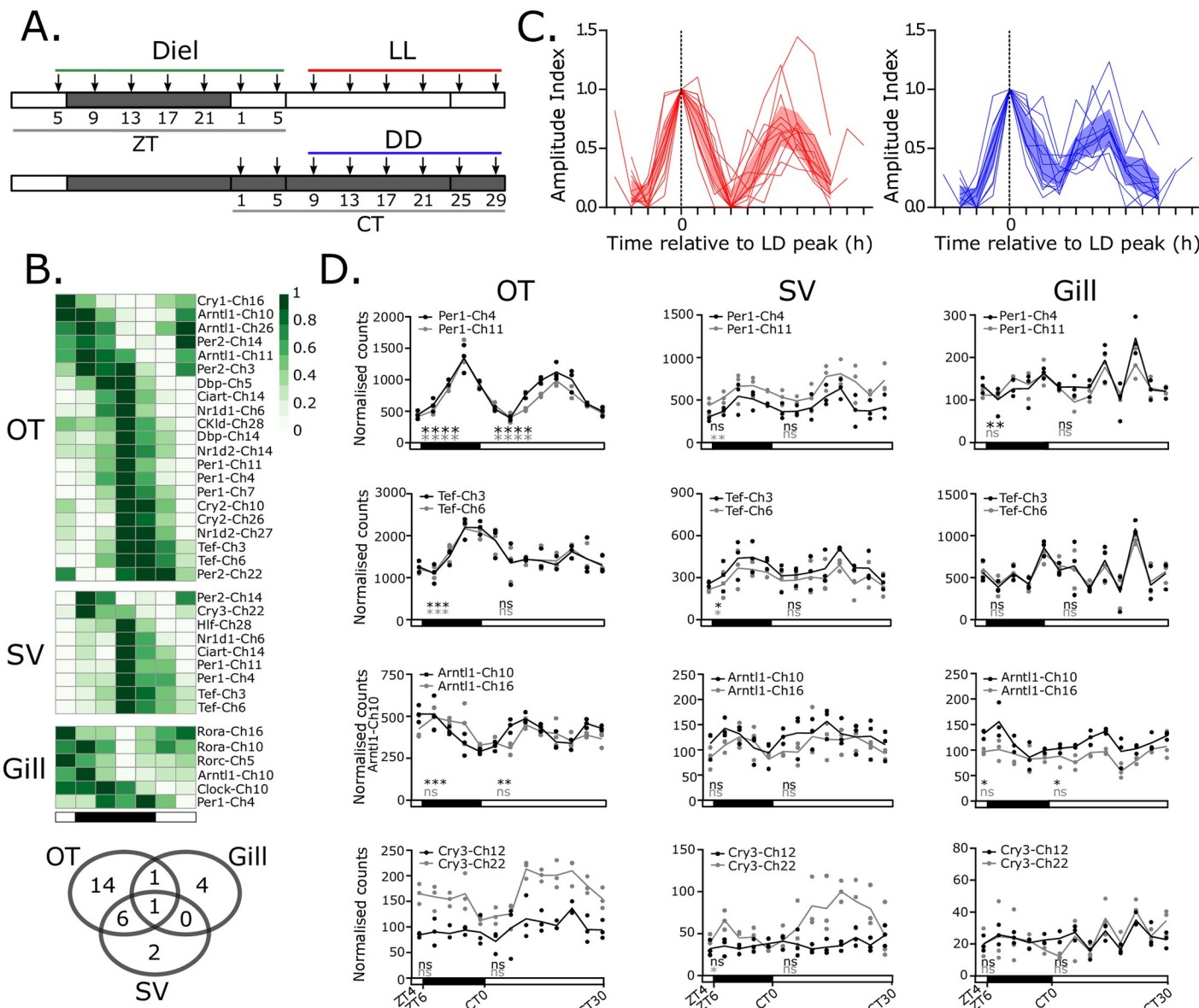

**Fig 2. The molecular clock network regulation in the Atlantic salmon.** A. Diel and circadian regulation experiment design, arrows indicate tissue collection points. Fish were maintained in a light dark cycle ("Diel"—short photoperiod 6 hours light: 18 dark) transitioning onto either constant light (LL) or constant dark (DD). Sampling in diel conditions was at a 4 hourly resolution starting from 5 hours after lights on (Zeitgeber time—ZT5) for 24 hours. The green line represents the time-points used in the statistical analyses of diel rhythmicity. Constant conditions are defined by circadian time (CT) which is relative to lights on in the preceding cycle. The assessment of DD or LL rhythmicity was made using samples from CT9 to CT29, indicated by the red and blue lines. N = 3 for all time-points. B. Heatmap displays all diel rhythmic genes in OT, SV and gill. Overlap between tissues is shown in the venn diagram. C. Peak-phase aligned LL and DD rhythmic genes in the OT. Shaded area shows 95% confidence limit. D. Example duplicate comparisons from OT, SV and gill. JKTcycle (adjP<0.05*, adjP<0.01**, adjP<0.001***, adjP<0.0001****).

## Circadian and light-regulated Ss4R pair expression differs between tissues but is highly similar within tissues

To test for circadian regulation of gene expression we collected samples from fish kept in a light dark cycle (diel), constant light (LL) and constant dark (DD) (Fig 2A, see Materials and methods, *Circadian experiment I*). To avoid unintended rhythmical stimuli (zeitgebers), fish were fasted from 48 hours before the first sampling point and temperature was held constant.

We focussed on three tissues with distinctive roles in salmonid physiology: the optic tectum (OT) of the brain, because it is linked to visual processing and is coupled directly and indirectly to light input [26–29]; the saccus vasculous (SV) because it has been proposed as a mediator of photoperiodic responses [30]; and the gill because it is essential for respiratory gas exchange, ion- and water balance [31]. We hypothesized that expression profiles of clock genes in these three tissues would differ reflecting tissue-specific differences in temporal metabolic demand. We analysed RNA transcript profiles using a bespoke NanoString CodeSet which could specifically identify 46 clock gene targets including 17 Ss4R pairs (S2 Table, S2A Fig and S2 Appendix).

In diel conditions, we identified 28 oscillating transcripts (JTK-cycle [32], adj.p<0.05, S1 Table) (Fig 2B). Of the three tissues studied, the OT showed by far the strongest oscillations in gene expression, both under diel and constant conditions (Fig 2B and 2C). For half of the genes identified, oscillation was only observed in the OT, and even for genes showing significant oscillation across tissues (e.g. Per1-Ch4) the amplitude of oscillation was clearly highest in the OT (Fig 2B and 2D).

In contrast, rhythmicity in both the SV and gill was much less robust. In the SV, only Nr1d1-Ch6 maintained rhythmicity and phase under DD, while in the gill only Arntl1-Ch10 maintained rhythmicity and phase under DD (S2B and S2C Fig). Hence robust circadian rhythmicity is a feature of the salmon brain, but gene expression rhythms are severely dampened in the peripheral tissues we studied.

Although differences in absolute expression levels were widely seen within Ss4R pairs—both across and within tissues, when comparing temporal dynamics of expression, within a given tissue they were typically similar (Fig 2D, S1 Table, S2 Appendix). This is exemplified by the almost superimposable expression patterns seen for the Per1-Ch4/11 pair (Fig 2D), and for the Tef-Ch3/6 pair (Fig 2D). Indeed, only two significant within-pair differences in expression profile were observed (non-linear regression p-value <0.01, S1 Table, S2D and S2E Fig): the Arntl1-Ch10/16 pair, with Arntl1-Ch10 showing more robust and higher amplitude rhythmicity than Arntl1-Ch16 in the OT (Fig 2D and S2D and S2E Fig), and the arrhythmic Cry3-Ch12/22 pair, with Cry3-Ch22 showing a light-induced increase in expression following transfer to LL in the SV (Fig 2D and S2E Fig). Interestingly, we do not observe light responses through Tef as reported in zebrafish [33].

## Regulatory divergence of clock gene ohnologues within a tissue during a developmental transition

The lack of circadian regulatory divergence among Ss4R pairs led us to consider whether retention of duplicates might be related to developmental changes in tissue function. One striking example of this in salmon is the transformation of gills during smoltification from a salt retaining, water excreting organ in freshwater to a salt excreting water retaining organ in seawater [34]. This developmental transition during the anadromous lifecycle relies on hormonally-driven changes in physiology, dependent on seasonally changing day-length (photoperiod) [34]. We therefore performed a photoperiod manipulation experiment, over 110 days, to assess the impact of photoperiod-dependent developmental changes in juvenile salmon (parr) (Fig 3A, see Materials and methods, *smoltification experiment*). This protocol produces a seawater-tolerant (smolt) phenotype within 4–6 weeks of return to LL (S3 Fig) (reviewed in: [34]).

We identified 30 clock genes showing significant changes in expression over the 110 days of the experiment (FDR<0.01, S1 Table, Fig 3B); 3 clock genes were undetectable by RNAseq, while a further 28 were present but did not change significantly over time. Amongst the

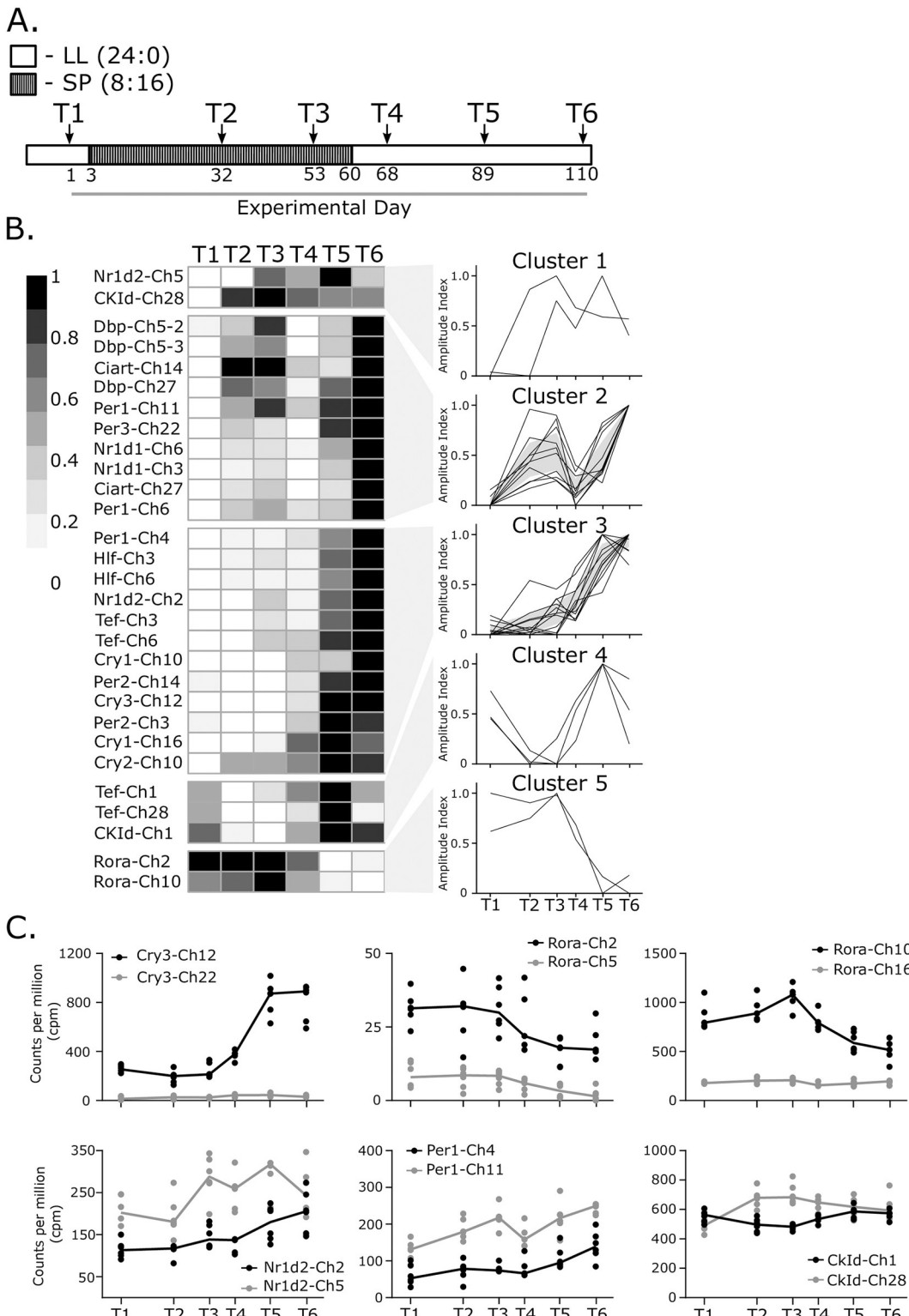

**Fig 3. The molecular clock network shows ohnologue specific differences during a photoperiodically driven developmental transition.** A. Photoperiodic gene expression experimental design. LL—constant light, SP—short photoperiod. Time-points for sampling (n = 6) indicated by the arrows, sampling was conducted in mid-light phase, time scale shown is the number of experimental days. B. Expression heatmap of significantly photoperiodic (FDR<0.01) clock genes in the gill. Significant genes cluster into five distinct expression patterns. Individual profiles are represented. Shaded area shows 95% confidence limit where applicable. C. RNAseq counts per million profiles for divergently regulated ohnologue pairs.

differentially regulated genes we found 9 Ss4R pairs (18 genes) where both copies within a pair showed smoltification-associated regulation, while for 3 pairs only one copy of the pair showed significant changes in expression during smoltification.

Gene expression correlation cluster analysis of the differentially expressed genes revealed 5 distinct patterns of expression over the experiment indicating that different regulatory pathways are directing the expression of particular clock genes during smoltification (Fig 3B). Based upon this analysis we identified 6 Ss4R pairs within which evidence of developmental regulatory divergence could be observed (Fig 3C). For 3 of these Ss4R pairs only one member showed a developmental change in expression (Fig 3C, Cry3-Ch12/22, Rora-Ch2/5, Rora-Ch10/16), while for the 3 remaining Ss4R pairs the dynamics of expression, as defined by cluster analysis, differed within the pairing (Fig 3C, Nr1d2-Ch2/5, Per1a-Ch4/11, CKIδ-Ch1/28). Therefore we see strong developmental regulation of clock genes in the gill over 110 days, contrasting with the lack of circadian regulation.

## Glucocorticoid signaling induces clock ohnologue expression and accounts for regulatory divergence observed in the Ss4R pair Tef-Ch3/6

While glucocorticoids play a major role in the circadian organization of mammals (reviewed in: [35]), the evidence for an analogous role in fish is unclear (References summarized in: S3 Table). Nevertheless, cortisol is a major hormonal regulator of smoltification in Atlantic salmon, steadily rising during this photoperiod-driven seasonal process [36]. We collected blood samples from fish kept in a light dark cycle (LD—6:18) and in constant conditions (LL or DD) and found no evidence of diel or circadian rhythmicity in cortisol secretion (Fig 4A, see Materials and methods, *Circadian experiment II*, S4 Table) along with weak or absent peripheral tissue clock gene circadian oscillation (S2B and S2C Fig). We hypothesized that the changes in gene expression observed in Fig 3 may be due to seasonally increasing cortisol during smoltification, therefore if we induced cortisol through a simple stress test we may induce the same clock genes seen during smoltification.

To test this we conducted a 24 hour seawater (SW) challenge test in freshwater-adapted fish (Fig 4B, see Materials and methods, *Smoltification experiment*) eliciting an osmotic stress-mediated increase in cortisol secretion (Fig 4C). Gills were collected from SW and fresh water (FW) groups. We identified 15 clock genes showing significant changes in expression in response to SW by RNAseq (FDR<0.01, S1 Table, Fig 4D). Importantly, 87% of acutely SW-responsive clock genes (13/15) also change over the chronic developmental time-scales of smoltification (Fig 3, S1 Table). Amongst the SW responsive genes we found 3 Ss4R pairs (6 genes) where both copies responded to SW, and 6 pairs where only one of the pair changed expression in SW. To assess regulatory divergence within these 9 pairs we plotted the fold change in response to SW for each copy of the pair and determined that 5 of the pairs showed significant regulatory divergence (two-way ANOVA <0.01, S1 Table, Fig 4E).

To further examine if glucocorticoid signaling, via cortisol, was responsible for the induction of clock genes in the gill we used transcription factor binding site analysis [37] on clock genes induced by SW (15 genes) compared to 43 clock genes that were SW-insensitive. SW-induced circadian genes promoters were highly enriched for HSF1 (heat shock factor 1), FOXO1 (forkhead box O1), MAX1 (myc-associated factor X1) and glucocorticoid receptor response elements (GR) (Fig 4F, S5 Table). Smoltification and responses to SW-exposure are coordinated by multiple endocrine factors including cortisol, growth hormone (GH) and IGF-1 [38,39]. HSF-1 and FOXO-1 elements are regulated by IGF1 signaling, during stress, cellular metabolism and development [40–44]. Furthermore, the enrichment of GR implicates non-

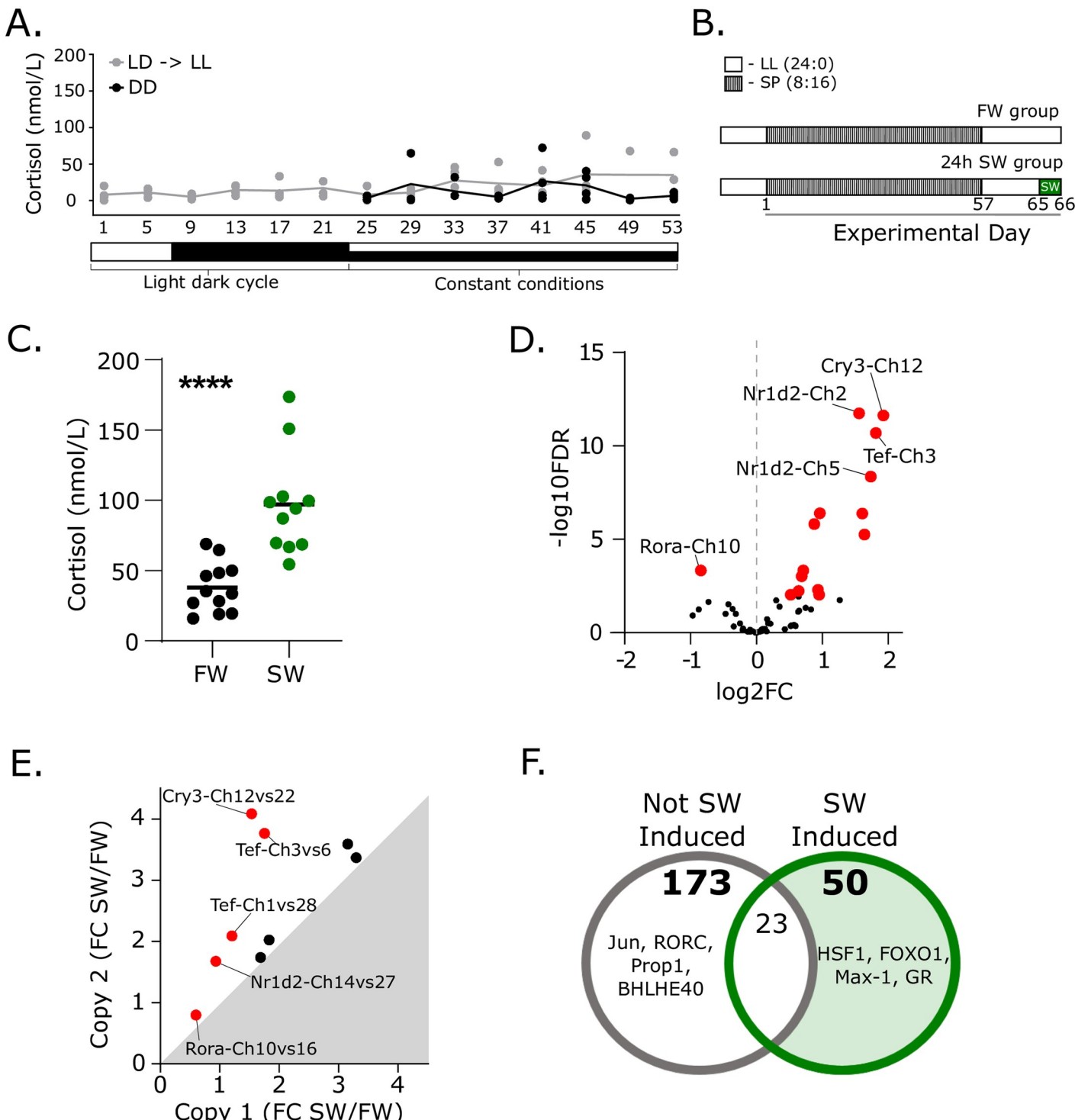

**Fig 4. Stress response implicates glucocorticoid receptor signaling in clock ohnologue regulation.** A. Diel and circadian profile of plasma cortisol (n = 4). Time axis is given in continuous hours since the start of the experiment, therefore 1 to 21 represent zeitgeber time (ZT) and 25 onwards is equivalent to circadian time (CT)1 to CT29. Due to the sampling protocol time-points 1 and 5 are replotted from time-points 25 and 29. B. Sea-water stress experiment design. LL—constant light, SP—short photoperiod, SW- sea-water challenge. C. Plasma cortisol concentration in blood plasma in sea-water stress experiment (n = 11–12). D. Volcano plot showing sea-water stress regulation of clock genes (n = 6). Significantly regulated transcripts (FDR<0.01) are shown in red. FC—fold change. E. Differential sea-water stress regulation of ohnologue pairs. Significantly different pairs (Analysis of genes where one or both genes are significantly regulated by seawater (FDR<0.01), then submitted to a two-way ANOVA, with sea-water regulation and interaction, p<0.05) are shown in red. F. Predicted transcription factor promoter binding analysis. Both sea-water induced and not-induced gene cohorts were analysed. 50 motifs were specific to the sea-water induced cohort. The top four motifs in each group are displayed.

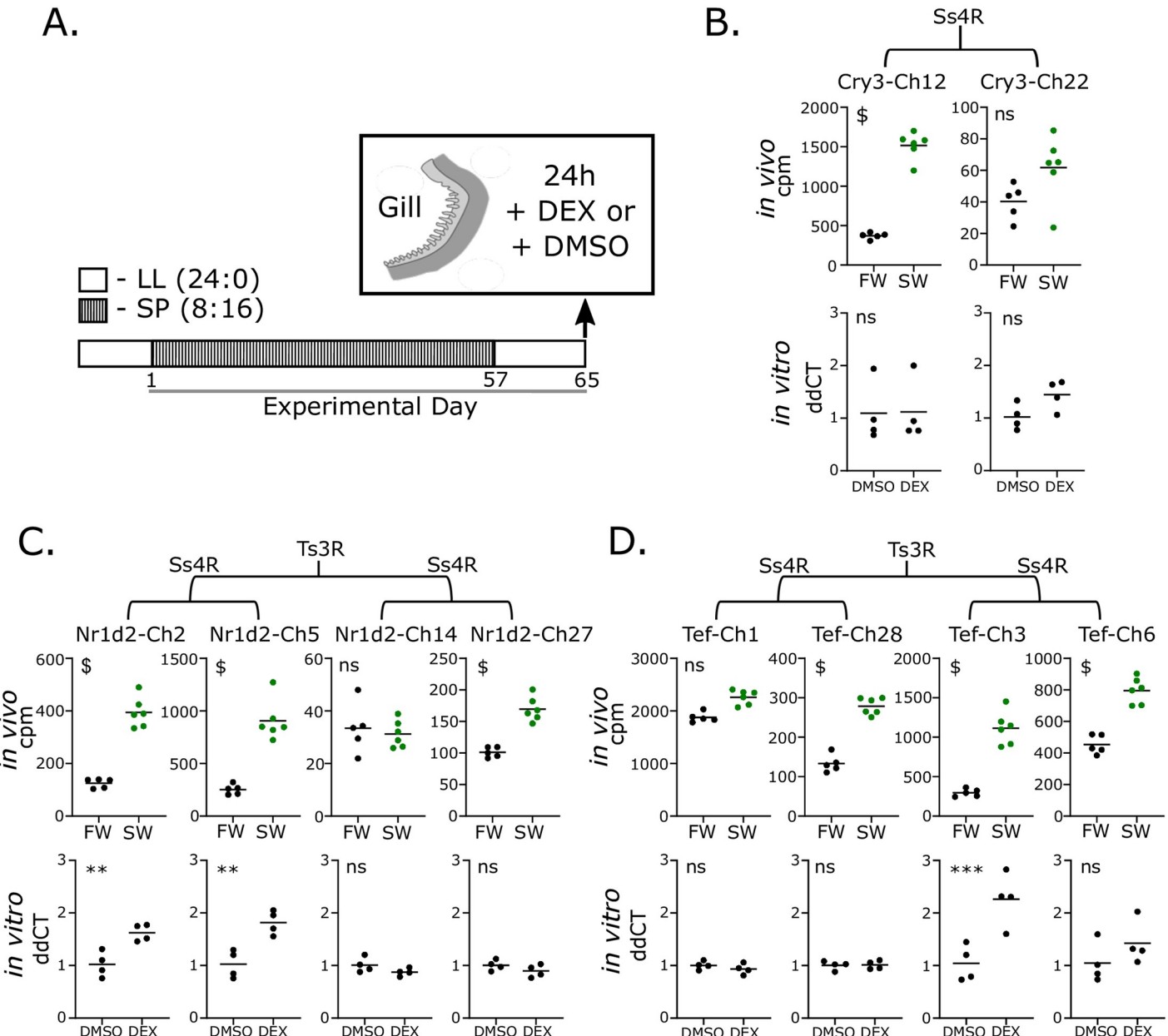

**Fig 5. In-vitro validation of glucocorticoid stimulated clock ohnologue expression.** A. Dexamethasone-dependent transcript regulation experimental design. After photoperiod manipulation *in vivo*, whole gill arches (n = 4) were removed and treated for 24 hours with the glucocorticoid receptor agonist dexamethasone (DEX), or dimethlysulphoxide (DMSO; vehicle control). B. Comparative regulation of Cry3-Ch12/22 taken from *in vivo* sea-water stress experiment and *in vitro* dexamethasone treatment. C. As in B for Nr1d2-Ch2/5 & Nr1d2-Ch14/27. D. As in B for Tef-Ch1/28 & Tef-Ch3/6. Braches indicate phylogenetic relationship between gene sets.

circadian glucocorticoid signaling in the induction of clock genes in the gill during smoltification and stress.

Next we wanted to experimentally test the *in-silico* association with glucocorticoid signaling and ask whether differential sensitivity to glucocorticoid signaling might account for the differential regulation observed within Ss4R pairs. We treated isolated gill arches with dexamethasone (DEX; a GR agonist) for 24 hours and then measured the gene expression of the 5 seawater stress regulated Ss4R pairs using qPCR (Fig 5A, S6 Table). We validated the

experiment by assessment of a positive control gene: *Tsc22d3* (*Gilz*) [45](S4 Fig). We found that whilst some of the genes are glucocorticoid sensitive (3/10), supporting the *in-silico* association, the majority are not induced by glucocorticoids (Fig 5B–5D). However, we do demonstrate divergence in the regulation of the Tef-Ch3/6 pair in response to glucocorticoids (Fig 5D).

## Discussion

Divergent regulation of gene expression is a major contributor to the evolution of novel adaptations and species diversity [46,47]. Our analysis of the circadian clock network in the Atlantic salmon reveals clear evidence of diversified expression regulation among the many retained copies of canonical clock genes that have arisen through successive WGD events. Diversified regulation is seen in tissue-specific expression patterns (Fig 1), and in response to developmental and environmental stimuli (Figs 3, 4 and 5). Ultimately, these differences likely reflect the uneven influence of diel factors on metabolism and cellular processes in central and peripheral tissues in a cold-blooded fish. Altered sensitivity to GR-signalling, likely mediated through the cortisol axis, emerges as one proximate cause of differences in tissue-specific expression dynamics (Figs 4 and 5). Overall, this analysis emphasises the potential for a richly diversified clock gene network to serve non-circadian functions in vertebrate groups with complex genomes.

A striking contrast emerges between clock gene expression regulation in the OT and the gill. The OT is a brain site which is coupled to light input both through retinal afferents, photoreceptor expression, and indirect melatonin-mediated input via OT melatonin receptors [26–29]. In contrast, the gill is directly exposed to the water environment and continuously handles the osmotic and energetic challenges that this presents [48–50]. Thus while both tissues are highly metabolically active and heavily reliant on ATPase activity to maintain function, the principal environmental influence for the OT is rhythmic light dark input, while for the gill it is continuous osmotic challenge.

Given these differences in tissue function, it is not surprising to observe different complements of clock genes in the two tissues and dramatic differences in temporal dynamics (Fig 2). In the diel-sensitive OT a dominant diel / circadian influence on expression is seen, while in the gill this is a much weaker influence. Nonetheless, clock gene expression in the gill is dynamically regulated, both over the chronic developmental time-scales of smoltification, and acutely in response to osmotic stress. The finding that the complement of genes showing highly sensitive changes in expression in the gill is largely non-overlapping with that under light / circadian control in the OT is clear evidence for diverged expression regulation within the circadian network.

Previous work has identified several Atlantic salmon clock genes that cycle in diel conditions [51–53]. While confirming these earlier findings, our analysis goes far beyond them by showing how the entire clock network is regulated under diel and constant conditions in several tissues. We show that only the OT shows persistent rhythmicity under constant conditions, not the SV or gill (Fig 2). These results sit in contrast to the zebrafish circadian system in which tissue autonomous clock gene oscillation are seen in all tissues investigated [54]. Specific elements of the zebrafish clock appear to be fundamentally different to the Atlantic salmon clock. For example zebrafish Cry1a is acutely light-induced through a D-box in it's promoter, via reactive oxygen species (ROS) and the JNK/MAPK pathway [55]. Importantly this light-mediated ROS inducibility of zebrafish Cry1a appears to be absent in cave fish and mammals [55]. Zebrafish Cry1a has also been reported to pause the circadian mechanism under LL [9], whereas the closest salmon orthologues to zebrafish Cry1a, Cry1-Ch10 and Cry1-Ch16, are

not light-induced and show persistent rhythmicity in the OT under LL (S1 Table). Instead the salmon orthologues to zebrafish Cry3b, Cry3-Ch22 and Cry3-Ch12, are acutely induced by light and salt, respectively. Light induction of Zebrafish Tefa [33] is not seen in the closest salmon orthologues Tef-Ch1 and Tef-Ch28, however, the zebrafish Tefb orthologues, Tef-Ch3 and Tef-Ch6 are seawater induced. In zebrafish, Tefb transcription is induced in response to light and UV exposure and then mediates the expression of other clock genes, initiating a DNA damage repair cascade [56]. This suggests a zebrafish-specific adaptation to UV light exposure, potentially related to spawning environment [56]. In our study we found no enrichment for genes associated with oxidative stress in either the seawater challenge or photoperiodic treatment groups but the photoperiod-driven smoltification induction of all four Tef genes is noteworthy, as is the seawater specific induction of the Ss4r pair Tef-Ch3 and Ch6, and the apparent regulatory divergence of these duplicates in glucocorticoid responsiveness. These species differences indicate plasticity of the circadian clock in response to different environmental stressors depending on life history.

Within the Atlantic salmon the distinct tissue-specific expression dynamics we observe and the smoltification and stress-related induction of clock genes focuses attention on cortisol [34,35,57]. In mammals cortisol is a major player in circadian organization, acting as an internal zeitgeber through which the hypothalamic–pituitary–adrenal (HPA) axis can coordinate daily changes in tissue activity throughout the organism (reviewed in: [35]). Because the teleost hypothalamic–pituitary–interrenal (HPI) axis is functionally analogous to the HPA axis [57], there has been a widespread assumption that cortisol plays a similar circadian coordination function in teleosts [58], although definitive evidence for this is lacking (S3 Table) [59]. Indeed our study implies that it is unlikely that cortisol plays a circadian role in salmonids–we found no evidence for circadian or even diel changes in cortisol secretion in Atlantic salmon (Fig 4A), and GREs are less enriched in circadian oscillating clock genes than in non-oscillating, development- and SW-sensitive clock genes (Fig 4F). Hence while our data strongly implicate cortisol in the dynamic expression of a subset of Atlantic salmon clock genes, this seems to have nothing to do with circadian function *per se*, and more to do with a role for these genes in non-circadian influences of the HPI axis. It is interesting to speculate that this shifting relationship between glucocorticoids and clock genes could be a contributory factor for the evolution of anadromy and the regulation of its seasonal timing. Furthermore, this finding raises interesting questions about the ubiquity and evolutionary origins of mammal-like coupling between the HPA axis and circadian function.

While evidence of functional divergence in ancient duplicated cryptochromes, and, sequence divergence of Ts3R cryptochromes in zebrafish have been observed [60] we expected that Ss4R generated significant genetic complexity in the salmonid circadian clock and therefore asked how expression patterns diverged within Ss4R pairs. Intriguingly, the answer to this question was highly dependent upon context. We saw many examples of pronounced within-Ss4R pair differences in terms of tissue-specific expression, and some 50% of identifiable Ss4R pairs showed within-pair divergence in expression during smoltification. But within-pair divergence in daily expression patterns was hardly observed. Why might this be so? We suggest this may reflect a difference in the way that selection pressures have operated on promoter regions to, on the one hand modulate tissue-specific expression and, on the other daily temporal regulation. According to this view duplication would confer freedom to diverge, thereby meeting differing tissue-specific requirements. Conversely, the daily temporal patterning may be so fundamental to cell function that any mutations leading to deviation from the ancestral dynamics were strongly selected against. This conjecture will require detailed analysis of regions of promoter conservation / divergence among Ss4R pairs.

## Materials and methods

### Ethics statement

Fish handling and euthanasia was performed by competent persons and in accordance with the European Union Regulations concerning the protection and welfare of experimental animals (European directive 91/492/CCE). The experiment was approved by the Norwegian Committee on Ethics in Animal Experimentation (ID 3630).

### Evolutionary analysis

To identify gene orthologs and ohnologs we generated protein sequence homology based orthogroups using the Orthofinder pipeline [61]. For each orthogroup we used the resulting protein tree topology to manually annotate pairs of salmon ohnologs based on the following criteria: (i) salmon ohnologs should form a monophyletic clade only containing genes from other salmonid species, (ii) this monophyletic salmonid clade must have Northern pike as the sister group, and (iii) putative ohnolog pairs had to be conserved in minimum one other salmonid species. Finally, we only retained putative ohnolog if their genomic positions were defined as syntenic regions originating from the Ss4R as defined in Lien *et al.* [22]. Maximum likelihood gene trees with bootstrap values for all genes included in the study are presented in S1 Appendix. These trees include the following species: human, mouse, spotted gar, zebrafish, stickleback, medaka, northern pike, grayling, danube salmon, arctic charr, coho salmon, rainbow trout, Atlantic salmon. The relevant scripts, input files and resulting trees are also available here. For quick reference S1 Table presents the salmon gene loci and their corresponding orthologues in Zebrafish, spotted gar, medaka and mouse.

### Multi-tissue analysis

Publically available data was used to assess the multi-tissue expression in the Atlantic salmon (one adult male, kept in freshwater), these data can be found in the NCBI Sequence Read Archive (SRA): PRJNA72713 and PRJNA260929.

### Animal husbandry

Juvenile Atlantic salmon (*Salmo salar*, Linnaeus, 1758) of the Aquagene commercial strand (Trondheim, Norway) were used in all experiments. Fish were held under constant light (LL; >200 lux), at 10˚C from hatching onwards, and kept in 500 L tanks from first feeding. The fish were approximately 7 months old when the experiments were initiated. Up until that time the fish had been feed continuously with pelleted salmon feed (Skretting, Stavanger, Norway), from automatic feeders.

### Circadian experiment I

Fish were maintained in 500L freshwater were transferred from LL to a short photoperiod (SP; 6L:18D) light schedule for 8 weeks before the start of the experiment. Short photoperiods provide a strong zeitgeber (timing signal) entraining the fish to a light dark cycle. In order to test for a circadian rhythm it is necessary to remove any zeitgebers, light, temperature and food are all potential zeitgebers which we controlled for. Two weeks before sampling, fish were distributed to two separate 150L tanks and water temperature was maintained at 14˚C (±0.5). Fish were fasted for 48 hours prior to the experiment and throughout the sampling. To identify genes which were driven by an endogenous circadian rhythm we sampled under diel (SP; 6L:18D), constant light (LL) and constant dark conditions (DD). Sampling was at 4 hour resolution starting at ZT5 (zeitgeber time; time since lights on) in diel conditions (n = 3 per time

point, 7 time-points), transitioning onto either LL or DD. In LL sampling was from CT9 (circadian time) at a 4 hour resolution (n = 3 per time-point, 6 time-points) because ZT 1 and 5 in this design can also be defined as CT1 and 5. In DD, samples were collected from CT1 at a 4 hour resolution (n = 3 per timepoint, 8 time-points) (Fig 2A). For the statistical analysis ZT5 to ZT5 was used for diel conditions, and CT9 to CT29 for constant conditions. Collections during the dark phase were conducted under dim red light. During sampling fish were netted out and euthanized by an overdose of benzocaine (150ppm). Weight and length were recorded and no significant variation noted. Optic tectum, gill and saccus vasculous were dissected and snap frozen on dry ice. RNA was extracted for subsequent nanostring profiling.

## Circadian experiment II

Fish were maintained in 500L freshwater were transferred from LL to a short photoperiod (SP; 6L:18D) light schedule for 20 weeks before the start of the experiment. Two weeks before sampling, fish were distributed to two separate 150L tanks. Fish were fasted for 48 hours prior to the experiment and throughout the sampling. Temperature was maintained at an average of 8.5˚C ±1. Sampling was at a 4 hour resolution starting one hour after lights on (ZT1), maintaining the fish under diel conditions (SP; 6L:18D) for 24 hours and then switching to either LL or DD, and sampling at 4h resolution for a further 29 hours (n = 4 per time-point) (Fig 4A). Collections during the dark phase were conducted under dim red light. During sampling fish were netted out and euthanized by an overdose of benzocaine (150ppm). Weight and length were recorded and no significant variation noted. Blood was collected at each timepoint from the caudal vein in heparinized vacutainers. Blood samples were centrifuged at 500 x g for 15 min to collect plasma for subsequent hormone analysis.

## Smoltification experiment and seawater tests

Using a standard aquaculture and research method we used photoperiod to induce the developmental transition of juvenile salmon (parr) to seawater prepared fish (smolts). This method requires switching parr on constant light to short photoperiod (SP; 6L:18D) for at least 8 weeks before returning the fish to constant light where over 4 weeks they gain osmo-regulatory capacity in seawater (Fig 3A). Fish were maintained in 150L freshwater tanks at an average 8.5˚C ±1 and were transferred from LL to SP (8L:16:D) for 8 weeks before return to LL for 7 weeks. Fish were sampled at six time-points in fresh-water shown in Fig 3A (T1-T6, n = 6 per time-point), these time-points are designed to capture the photoperiodically induced transition from parr to smolt. Fish were fasted for 48 hours prior to sampling. Gills were collected into RNAlater (Sigma-Aldrich, St. Louis, Missouri, USA), storing at 4˚C for 24 hours, before being transferred to -80˚C. RNA was extracted and used for RNA-seq (Data shown in Fig 3).

To validate the photoperiod protocol, seawater challenge tests were conducted on a randomly selected subgroup that were transferred to a 100L tank supplied with seawater (34‰ salinity) for 24 hours. Fish were netted out after 24 hours and euthanized by an overdose of benzocaine (150ppm). Blood samples were collected from the caudal vein in heparinized vacutainers and plasma was collected and stored at -20˚C. To test the osmolality of blood thawed plasma samples were analysed for osmolyte content using a Fiske One-Ten Osmometer (Fiske Associates, Massachusetts, USA, ± 4 mOsm kg-1). These tests were conducted from T2 to T6 (n = 12 per time-point), and confirm that from T5 (29 days after the return to LL) the fish had osmo-regulatory capacity in seawater (S3 Fig). Gills were collected at each time-point after seawater challenge (n = 6 per time-point), RNA was extracted and RNA-seq performed as above. All data are available from the European nucleotide archive under project number: PRJEB34224. This study presents data from the freshwater groups at all time-points (Fig 3)

and seawater challenged fish at T4 only (Fig 4B). T4 is 8 days after return to LL and therefore these fish do not have osmo-regulatory capacity in seawater and are used as a stress test comparison to fish from the same time-point kept in freshwater (Fig 4B–4F).

### In-vitro Gill Culture

Juvenile Atlantic Salmon were prepared as in the smoltification experiment, sampling at the equivalent of T4 (8 days after return to LL) (Fig 5A). Following euthanasia whole gill arches were rapidly dissected (biological replicates, n = 4 per treatment), excess mucus was removed by careful blotting onto tissue paper before the arches were transferred individually into 50 ml of pre-prepared control or treatment media. The prepared media consisted of Leibovitz L-15 (Lonza) supplemented with non-essential amino acids (1%, 100x Lonza), sodium-pyruvate (1%, 100x Lonza), 0.05 mg/ml gentamycin (Sigma) and 20% fetal bovine serum (FBS, sigma). The experimental group was supplemented with 0.1M dexamethasone diluted in DMSO (dimethyl-sulphoxide, Sigma) to a final concentration of 0.1μM. The control group contained an equivalent concentration of DMSO (0.1%). The excised gill arches were incubated for 24 hours at 4˚C, gill filaments were removed with a scalpel and snap frozen on dry ice before being stored at -80˚C. RNA was extracted for qPCR analysis.

### RNA extraction

RNA extraction for RNAseq was performed using a TRIzol-based method (Invitrogen, Thermo Fisher, Waltham, Massachusetts, USA), and in accordance with the manufacturers recommendation. Resulting RNA concentrations and quality were checked using a NanoDrop spectrophotometer (NanoDrop Technologies, Wilmington, DE, USA). RNA was stored at -80˚C.

For nanostring and qPCR, SVs were extracted using QIAgen RNeasy micro kit, OT and gill tissues were extracted using QIAgen RNeasy mini kit according to the manufacturers instructions. RNA concentration was quantified and quality confirmed using the Experion Automated Electrophoresis System (BioRad).

### Nanostring

Custom nanostring codesets were designed by Nanostring Technologies Inc. using the Atlantic Salmon reference genome (Cigene), accession numbers and target sequences are shown in S2 Table. Codesets were processed by the Univerisity of Manchester Genomic Technologies Core Facility. This technology is based on the use of fluorescent barcoded probes which bind specifically to the target molecule. Importantly these barcodes should only bind one at a time to each target molecule, therefore the number of fluorescent barcodes reflects the number of RNA molecules of your target gene. A spike control with a known number of RNA molecules was also used to normalise across samples and runs. Therefore the units are counts normalised to spike-in positive controls. Data was processed using nSolver 4.0 software (Nanostring). Data can be accessed on GEO under the project identifier GSE146530.

### Transcriptome sequencing and assembly

Libraries were prepared using TruSeq Stranded mRNA HS kit (Illumina, San Diego, California, USA). Mean library length was determined using the 2100 Bioanalyzer with the DNA 1000 kit (Agilent Technologies, Santa Clara, California, USA). Library concentrations was determined using the Qubit BR kit (Thermo Scientific, Waltham, Massachusetts, USA). Samples were barcoded with Illumina unique indexes. The Illumina HiSeq 2500 was used to

perform single-end 100-bp sequencing of samples at the Norwegian Sequencing Centre (University of Oslo, Oslo, Norway).

Cutadapt (ver. 1.8.1) was used for removal of sequencing adapters and trimming of low quality bases (parameters–q 20, -O 8 -minimum-length 40). Quality control was performed with FastQC software. Reads were mapped onto the references genome using STAR software (ver. 2.4.2a). Read counts for annotated genes were generated using the HTSEQ-count software (ver. 0.6.1p1).

All RNAseq data for the smoltification experiment is available in the European nucleotide archive under project number: PRJEB34224.

## Analysis of differentially expressed genes

Analysis of differential gene expression was performed with package edgeR (ver. 3.14.0) using R (ver. 3.4.2) and RStudio (ver. 1.0.153). Prior to analysis of differential expression, the raw counts were filtered, setting an expression level threshold of a minimum of one count per million reads (cpm) in five or more libraries, resulting in a list of 33 951 expressed genes. The counts were scaled by applying trimmed means of M-values (TMM) scaling. Exact tests were then performed to find genes that were differentially expressed between FW-kept and 24-hour SW challenged fish. An ANOVA-like test was performed to find genes that were differentially expressed over T1-T6 FW time-points. The test results were filtered for a false discovery rate (FDR) to be less than 0.01 to identify significantly differentially expressed genes. Clustering analysis was performed using Pearson correlation.

Heatmaps were generated in R using custom scripts for pheatmap. Transcription factor binding site analysis was conducted using SalmotifDB [37].

## qPCR

cDNA was synthesised from sample total RNA using high capacity RNA to cDNA kit (Applied Biosystems). qPCR was performed using GoTaq Master Mix (Promega) and a 96 well thermal cycler (Applied Biosystems). Relative gene expression was quantified by the ΔΔCT method using *Ef1a* as reference gene. Primer sequences are listed in S2 Table and source data is in S6 Table.

## Hormone assays

Cortisol ELISA assays were performed by Stockgrand (UK). Source data in S4 Table.

## Statistical analyses

RNAseq analysis is detailed above. Mean difference comparisons were carried out using Student's t-test (two-sided, unpaired), two-way ANOVA with post hoc tests as appropriate (Graphpad Prism 8.1.2). The expression divergence index (EDI) index was calculated as follows: $EDI = abs(log2[Gene1/Gene2])$.

The R package JTK cycle was used to assess rhythmicity of transcripts under LD and constant light or dark conditions [32]. For statistical comparison of gene expression between ohnologue pairs in the circadian experiment, expression was normalized to group mean then best fit sixth-order centered polynomial curves were generated by non-linear regression analysis and shared characteristics tested with extra sum of squares F test (Graphpad Prism 8.0).

## Supporting information

**S1 Fig. Tissue specific expression of clock ohnologues.** A. PCA plot showing the relative tissue differences when considering clock ohnologue expression. B. Heatmap showing the tissue specific expression of clock ohnologues.
(TIF)

**S2 Fig. Nanostring clock gene expression and circadian phase aligned plots.** A. Heatmap showing the mesor expression for each clock ohnologue in three tissues. Grey indicates the gene is not expressed. B. Phase aligned plots for the gill. C. Phase aligned plot for the SV. D. Arntl1-Ch10/16 comparison: plot of non-linear regression using a sixth-order centered polynomial to fit the data and compare individual curves. P-value is the result of extra sum-of-squares F test. E. As above for Cry3-Ch12/22.
(TIF)

**S3 Fig. Osmoregulatory capacity during the smoltification experiment.** Osmolality (mOsm kg-1) is displayed for fish in freshwater (FW—blue) and seawater (SW—green) (n = 6). This plot show osmoregulatory capacity develops by the two latest timepoints (T5 and T6).
(TIF)

**S4 Fig. Gene expression of Tsc22d3-Ch3, a positive control gene for DEX treatment.** A. Gene expression of Tsc22dd3-Ch3 *in vivo* sea-water stress experiment (RNAseq counts per million (cpm)) and B. *in vitro* dexamethasone treatment (qPCR).
(TIF)

**S1 Table. Clock genes identified in Atlantic Salmon, orthogroups, duplicates, significances for the circadian, smoltification and seawater challenge experiments.**
(XLSX)

**S2 Table. Nanostring codeset design and qPCR primers.**
(XLSX)

**S3 Table. Summary of previous studies measuring cortisol in fish.**
(XLSX)

**S4 Table. Cortisol source data for Fig 4A.**
(XLSX)

**S5 Table. SalmotifDB results—Transcription factor binding site analysis.**
(XLSX)

**S6 Table. qPCR source data for Fig 5.**
(XLSX)

**S1 Appendix. Evolutionary gene trees for circadian clock genes.**
(PDF)

**S2 Appendix. Nanostring circadian profiles for all genes.**
(PDF)

## Acknowledgments

The authors thank all of the animal staff at Kårvik havbruksstasjonen for their expert care of the research animals.

## Author Contributions

**Conceptualization:** Alexander C. West, David G. Hazlerigg, Shona H. Wood.

**Data curation:** Alexander C. West, Marianne Iversen, Simen R. Sandve, Shona H. Wood.

**Formal analysis:** Alexander C. West, Marianne Iversen, Even H. Jørgensen, Simen R. Sandve, Shona H. Wood.

**Funding acquisition:** David G. Hazlerigg, Shona H. Wood.

**Investigation:** Alexander C. West, Even H. Jørgensen, Shona H. Wood.

**Methodology:** Alexander C. West, Marianne Iversen, Even H. Jørgensen, Shona H. Wood.

**Project administration:** Even H. Jørgensen, Shona H. Wood.

**Resources:** David G. Hazlerigg, Shona H. Wood.

**Software:** Simen R. Sandve.

**Supervision:** Even H. Jørgensen, David G. Hazlerigg, Shona H. Wood.

**Validation:** Alexander C. West, Shona H. Wood.

**Visualization:** Alexander C. West, Simen R. Sandve, Shona H. Wood.

**Writing – original draft:** Shona H. Wood.

**Writing – review & editing:** Alexander C. West, Marianne Iversen, Even H. Jørgensen, Simen R. Sandve, David G. Hazlerigg, Shona H. Wood.

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
