## [Decision Letter · Decision Letter 0]

3 Jun 2020

Dear Dr Wood,

Thank you very much for submitting your Research Article entitled 'Diversified regulation of circadian clock gene expression following whole genome duplication' to PLOS Genetics. Your manuscript was fully evaluated at the editorial level and by independent peer reviewers. You will see that the reviews are mixed but all agree that your manuscript reports on a question that is intrinsically interesting to the clock field. Yet, they all have comments regarding the work, which you will need to address in order for the manuscript to be re-considered.  In addition, you will see that Reviewer 2 brings up the very important issue of whether this work is "appropriate for this journal". We understand that you cannot turn the manuscript into something that it is not; we are just asking you to do your best to present the work from a perspective that will make it more suitable to the readership of the journal. 

If you decide to revise the manuscript for further consideration at PLOS Genetics, please aim to resubmit within the next 60 days, unless it will take extra time to address the concerns of the reviewers, in which case we would appreciate an expected resubmission date by email to plosgenetics@plos.org.

[LINK]

We are sorry that we cannot be more positive about your manuscript at this stage. Please do not hesitate to contact us if you have any concerns or questions.

Yours sincerely,

John Ewer

Associate Editor

PLOS Genetics

Kirsten Bomblies

Section Editor: Evolution

PLOS Genetics

Reviewer's Responses to Questions

**Comments to the Authors:**

Reviewer #1: This manuscript by West and colleagues tackles a central issue that has arisen in the study of circadian clock function in non-mammalian vertebrates, notably fish species. This concerns the functional significance of multiple copies of different types of clock gene that have arisen as the result of whole genome duplication. Does this reflect the presence of considerable functional redundancy or alternatively, do the multiple gene copies adopt different specialized functions and thereby confer even greater flexibility and adaptability on the circadian timing system during evolution? Until now, this issue has not been convincingly resolved. The authors of this manuscript approach the problem by studying a fascinating species, the Atlantic salmon, where multiple genome duplication events have lead to the extreme situation of there being 61 clock gene counterparts. Furthermore, the most recent genome duplication event has been attributed to enabling the evolution of anadromy a physiologically challenging adaptation whereby the fish spend parts of their life cycle in fresh water and other parts in sea water.

The authors charaterize daily changes in expression of these various clock genes in the gills, the optic tectum and the saccus vasculous as well as over the course of development. In particular, they compare the expression patterns of pairs of clock gene „ohnologues“. The results indicate strong gene and tissue-specific divergence in expression patterns consistent with functional diversification. Furthermore, a link between cortisol secretion and non-circadian expression of a subset of the clock genes is observed in the gill, which is thought to constitute part of the adaptation to the transition from fresh to sea water.

This represents a valuable and fascinating piece of work that will boost the relevance of many studies which explore circadian clock gene function and regulation in non-mammalian vertebrate models as well as providing valuable new insight for studying the evolution of the physiological function of cortisol. The data are well presented, convincing and are interpreted in a reasonable way

I have one general point of criticism that concerns the use of purely mammalian (mouse-derived) data to guide the choice and interpretation of the clock genes under investigation. In particular, predominantly following the work and conclusions of Uli Schibler, the PAR domain transcription factors TEF, DBP and HLF are all considered as clock-related genes - serving as clock regulated factors that mediate feedback to the core clock loop mechanism. To my understanding, several publications have now shown that this view of PAR factor function is clearly not the case in fish. These transcription factors have been instead identified as mediating light and UV induced transcription of clock genes such as per2 and Cry1 as well as other types of gene involved in DNA damage repair. Interestingly, they also appear to serve as mediators of oxidative stress-induced gene expression (Pagano et al., Scientific reports 2018). I think the key point here is that D-box enhancers, which act as the regulatory targets of these PAR factors, are NOT clock regulated in fish. This important observation is completely ignored in the text and actually raises a more general point that would actually support the main conclusions of this manuscript, namely that clock gene function and regulation is inherently plastic during evolution, and so major changes in the role of particular sets of clock components may be entirely expected. The introduction and discussion sections should be adapted to include this missing relevant information.

A related point: I am not sure about the current understanding of the mechanisms involved in the osmotic stress response of the gills, but might the observed differential regulation of TEF genes during the response to osmotic stress support some role for oxidative stress in this process?.....

Other points emerging from the text:

Line 53: What do the authors mean by (co-)activator and (co-)repressor? There are transcription factors that directly bind to DNA sequences and co factors that interact with transcription factors and other parts of the transcription machinery but do not directly bind to DNA. So, it would be more accurate to mention transcription factors and cofactors or coregulators.

Line 131 and 147: Describing the option tectum as „highly light-responsive“ at this point in the text is a little misleading - specifically because of the existence of directly light responsive peripheral tissues in fish. This is more clearly explained later in the discussion (retinal input, expression of photoreceptors etc.), but I think to avoid confusion, it would make more sense to also clarify this point in the introduction section..

Line 217: „during in stress“ should be corrected to „during stress“.

Reviewer #2: The review is attached as a word document.

Reviewer #3: The authors have presented a detailed analysis and description of the clock gene network

paralogs present in Atlantic salmon. It describes the tissue distribution and expression

differences between 4R paralogs. I think that the analysis is thorough and well executed with

followup experiments.

The main findings of the study is that the salmonid lineage specific whole genome duplication has

provided additional members of clock gene families that have had the possibility to diversify

in expression between different tissues and also in specialise to possibly perform other

functions then regulation of the circadian clock.

GENERAL COMMENT

Is it common practice to use the suffix -ChXX for Atlantic salmon gene names? I suppose it is

referring to the chromosome the gene is located on. If it is not common practise add an

description on this in the manuscript as to not confuse the reader. Alternatively use the

gene name format common to what is used in atlantic salmon.

INTRODUCTION

on line 59, 77: it is today well established that these two rounds of WGD are common among

at least all jawed vertebrates and not only tetrapods as stated here.

on line 80: It is important to note the 3R WGD is shared by all teleost fish.

TISSUE EXPRESSION OF 4R PARALOGS

Did you consider looking at the expression of these genes in the pineal organ, if so why was it

excluded?

I couldn't find any information in the salmon genome paper from which you have used data

regardning tissue specific expression does not state time of day of collection or how many

individuals were used. Do you have any information about this, that could be added? Could

this affect the results presented you think?

CIRCADIAN EXPERIMENTS

In the different circadian experiments the short photoperiod light schedule were kept for

different number of weeks (8 and 20), why was it done like this?

In the method section i cant find the intervals and timepoints for sampling and why they

were chosen, i would recommend adding this to the materials and method section.

Im not sure i understand the figures showing this since not all times are added and its

not clear when the start time is. Figure 4A is much clearer in this regard.

In the materials and methods section I cant find information about the number of individuals

were sampled per experimental condition/time-point.

DISCUSSION

How does the data presented here for the atlantic salmon fit in the context of expression

data presented in the literature from other species of salmonids?

There seem to be publications regarding the expression of some of the clock genes in

north american salmonids.

Did you observe any functional specialisation between paralogs resulting from 3R in those

cases where you didn't find any differences between the 4R paralogs?

Has functional divergence of 3R duplicates been described for other teleost species?

FIGURES

Figure 1A: Why is there no root in this tree, and what does the strange branching between

coelacanth and terapods mean?

Figure 1C: What does the lines that are not for the clock genes represent in this figure?

Is it possible to have different colors for the lines representing different clock 4R paralogs?

Figure 2:

What are the normalised counts, which unit?

Why is there two scales in figure 2D for Arntl-ch10 Arntl-ch16 in optic tectum. Looking at the

graph it seems better to use the scale to the right for both.

Figure 3:

Looking at the timeline in A, i see that the sampling time intervals differ a lot. Why is

the intervals of equal lengths in B and C. This changes the look of the graphs quite a lot

and could lead to missinterpretation.

SUPPLEMENTARY FIGURES

S1 Appendix: For the phylogenetic trees i would suggest adding support values on the nodes.

This would help the reader interpret the results better.

**Have all data underlying the figures and results presented in the manuscript been provided?**

Reviewer #1: Yes

Reviewer #2: Yes

Reviewer #3: No: The data for qPCR and Cortisol assays are not provided

PLOS authors have the option to publish the peer review history of their article (what does this mean?). If published, this will include your full peer review and any attached files.

Reviewer #1: No

Reviewer #2: No

Reviewer #3: No

---

## [Decision Letter · Decision Letter 1]

25 Aug 2020

Dear Dr Wood,

Thank you very much for submitting your revised Research Article entitled 'Diversified regulation of circadian clock gene expression following whole genome duplication' to PLOS Genetics. Your manuscript was re-evaluated at the editorial level and by the reviewers of original version.

The reviewers appreciated the changes you made to address their comments and concerns. You will see that what remains are a couple of minor corrections, which we now ask you to do before we can consider your manuscript for acceptance. Your revisions should address the specific points made by each reviewer.

[LINK]

Yours sincerely,

John Ewer

Associate Editor

PLOS Genetics

Kirsten Bomblies

Section Editor: Evolution

PLOS Genetics

Reviewer's Responses to Questions

**Comments to the Authors:**

Reviewer #1: The revised version of the manuscript has been much improved as a result of addressing this reviewer‘s comments as well as those of the other reviewers. In particular, introducing a comprehensive overview of previous relevant work that has been based on the zebrafish has helped to better place the current study into a more general context. However, it is important that the authors now correct an error that has been introduced with the new text.

Page 15-16 line 277-301

The authors now make reference to the paper, Pagano et al (Scientific Reports, ref. 55) which they state illustrates how the D-box enhancer in the context of the Tef promoter, mediates transcriptional inducibility by light, UV and ROS. I have just checked the details of this work, and it seems that the D-box that is studied is in the context of the Cry1a promoter, NOT the tef promoter. The authors should correct their text accordingly.

Reviewer #2: Thank you for updating and improving your manuscript. I believe considerable improvements have been made.

Reviewer #3: I am happy with the answers i received from the authors regarding my comments on the manuscript.

I have one remaining minor comments on the revised version. In figure 1A: I think you missunderstood my comment regarding the branching of the coelacanth. It was placed in the correct position in the original figure but the branching was not the traditional bifurcating branching so i wondered why you choose that way to display the sarcopterygian clade (human, mouse), coealanth) while (coho salmon, rainbow trout), arctic charr) were displayed in the traditional way.

**Have all data underlying the figures and results presented in the manuscript been provided?**

Reviewer #1: Yes

Reviewer #2: Yes

Reviewer #3: Yes

PLOS authors have the option to publish the peer review history of their article (what does this mean?). If published, this will include your full peer review and any attached files.

Reviewer #1: No

Reviewer #2: No

Reviewer #3: No

---

## [Editor Report · Decision Letter 2]

6 Sep 2020

Dear Dr Wood,

We are pleased to inform you that your manuscript entitled "Diversified regulation of circadian clock gene expression following whole genome duplication" has been editorially accepted for publication in PLOS Genetics. Congratulations!

Yours sincerely,

John Ewer

Associate Editor

PLOS Genetics

Kirsten Bomblies

Section Editor: Evolution

PLOS Genetics

Comments from the reviewers (if applicable):

**Data Deposition**

http://datadryad.org/submit?journalID=pgenetics&manu=PGENETICS-D-20-00531R2

**Press Queries**

---

## [Editor Report · Acceptance letter]

1 Oct 2020

PGENETICS-D-20-00531R2 

Diversified regulation of circadian clock gene expression following whole genome duplication 

Dear Dr Wood, 

We are pleased to inform you that your manuscript entitled "Diversified regulation of circadian clock gene expression following whole genome duplication" has been formally accepted for publication in PLOS Genetics! Your manuscript is now with our production department and you will be notified of the publication date in due course.

With kind regards,

Jason Norris

PLOS Genetics

On behalf of:
